# ConQuist: Condition Number Aware Quantization for LLMs

## Abstract

Post-training quantization (PTQ) of large language models (LLMs) has emerged as a promising technique in reducing the computational cost at inference time. Uniformly quantizing all weights and activations to 4-bit significantly degrades performance, due to the high quantization error caused by outliers present in activations. To mitigate this issue, we propose *ConQuist*, a PTQ method leveraging mixed precision quantization based on the condition number of each layer. The condition number quantifies the sensitivity of a layer's output to small perturbations in its activations; hence, layers exhibiting high condition numbers are prone to high quantization error. ConQuist identifies layers with higher condition numbers and allocates them higher precision (e.g., 5-bit), while quantizing the rest to 4-bit. We also provide a theoretical foundation that relates activation sensitivity to the condition number. Furthermore, we have empirically shown that our proposed ConQuist outperforms uniform PTQ methods, achieving up to 20% lower perplexity on a variety of benchmarks.

## 1 Introduction

Large language models (LLMs) have demonstrated remarkable performance across a variety of natural language processing benchmarks (Hendrycks et al., 2021) and have further expanded their capabilities to multimodal domains (Huang et al., 2024; Team et al., 2023). In particular, GPT (Brown et al., 2020) and LLaMA (Touvron et al., 2023) families have notably contributed to the ongoing advancement of LLM. However, the rapid progress and extensive adoption of LLMs have substantially increased computational demands and memory requirements. State-of-the-art generative models, including OPT-175B and LLaMA-65B, typically require hundreds of gigabytes of GPU memory, often necessitating deployment across extensive multi-GPU infrastructures (Zhang et al., 2022). Such substantial computational and memory demands present significant challenges for practical applications, especially in environments with limited resources (Pope et al., 2023; Smith et al., 2022). The increasing resource burden has thus motivated extensive research into model compression techniques aimed at maintaining model performance while reducing computational complexity and memory usage. To mitigate these resource constraints, quantization has emerged as an essential method, offering effective and efficient compression by discretizing model parameters and activations into lower-precision formats, thereby significantly decreasing both storage requirements and inference costs.

Quantization methodologies are generally categorized into Quantization-Aware Training (QAT) and Post-Training Quantization (PTQ). These two approaches address the challenge of accuracy degradation through fundamentally distinct strategies. QAT integrates quantization directly into the model training process, simultaneously optimizing both model weights and input activations (Chen et al., 2024a). Although QAT has shown strong performance across numerous benchmarks, its dependence on extensive datasets and lengthy training schedules makes it computationally demanding, thereby limiting its feasibility for practical deployment. In contrast, PTQ provides a substantially more efficient alternative, as it estimates quantization parameters using only a small calibration dataset, eliminating the need for retraining the entire model (Yao et al., 2022; Dettmers et al., 2022). PTQ has emerged as an effective model compression technique, striking a balance between performance and computational resource usage. PTQ has demonstrated notable success in quantizing the weights of pre-trained LLMs. however, substantial accuracy degradation remains a significant issue, particularly in low-bit quantization scenarios ($\leq$ 8-bit) (Frantar et al., 2023).

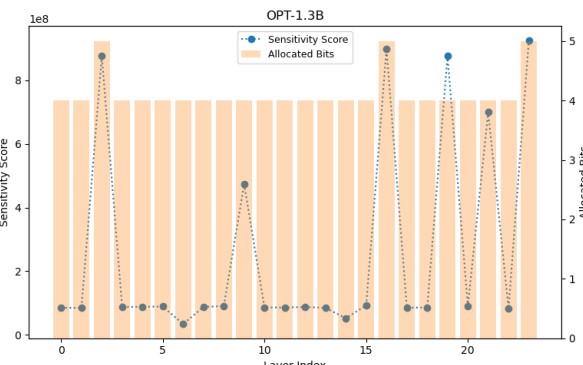

Figure 1: Layerwise bit allocation in the OPT-1.3B model, guided by sensitivity score for quantization.

In PTQ, uniformly quantizing of all layers to the same bit-width can result in a noticeable drop in performance, as different layers vary in their sensitivity to quantization. To address this challenge, recent research has introduced various mixed precision quantization methods aimed at preserving model performance (Li et al., 2025a; Zhao et al., 2024). Mixed precision quantization for LLMs, refers to a quantization strategy that assigns varying numerical precisions (bit-widths) to different parts or layers of LLMs (Ashkboos et al., 2023). Instead of uniformly quantizing the entire model to the same precision, mixed precision quantization allocates fewer bits (lower precision) to less sensitive layers, and more bits (higher precision) to layers whose accuracy strongly impacts overall performance. This approach effectively reduces model size and computational costs while minimizing accuracy loss.

Despite having several advantages, mixed precision quantization in LLMs faces two major challenges.

- **Ill-conditioning of activations:** Recent studies have shown that while weight matrices in LLMs tend to be high-rank or full-rank, activations often exhibit low-rank structures (Yu & Wu, 2023) and contain extreme outliers (Xiao et al., 2023). This combination leads to ill-conditioned activation distributions. Most existing mixed precision approaches primarily target weight quantization and overlook the complexities and instability introduced by the ill-conditioning of activations.

- **Dependence on second-order information:** Many mixed precision methods apply higher bit-widths to outlier or sensitive columns of weight matrices while keeping the rest at lower precision. Moreover, layer-wise mixed precision techniques typically require second-order information, such as the Hessian, to estimate the sensitivity of each layer to quantization (Dong et al., 2020). Computing such second-order statistics is computationally intensive and becomes impractical for LLMs due to their massive parameter counts.

To address these challenges, we propose *ConQuist*, a novel mixed precision quantization technique that leverages the condition number of layer-wise activations as a sensitivity metric to guide precision allocation. The condition number reflects how much the output can change in response to perturbations in the input. In quantization, the condition number of activations refers, the quantization error introduced in the weights. Specifically, it measures the amplification of quantization errors from weights to the layer's output. For small disturbances, a smaller condition number indicates that the layer's output varies less. We have observed that, only a few layers in LLMs have high activation condition numbers, implying that these layers are more sensitive to weight quantization than others, see figure 1.

ConQuist identifies layers with higher sensitivity score, characterized by larger condition numbers and assigns them a slightly higher precision (5-bit), while the remaining, less sensitive layers are quantized to 4-bit. This strategy keeps most of the model in lower precision, minimizing computational overhead, while selectively providing higher precision to critical weights, resulting in substantial performance improvements. We provide theoretical justification for using the condition

number as a sensitivity score, showing that the relative variance of quantization error is bounded by the condition number of activations. Unlike other sensitivity based methods that depend on second-order information (such as Hessian), our approach only requires access to activations from each layer, making it more efficient and scalable.

Integrated with GPTQ (Frantar et al., 2022), ConQuist improves the generalization capabilities of quantized large language models. Empirical results show that, when quantizing to 4-bit precision with just 10% of layers at 5-bit, ConQuist achieves similar performance same as full precision on large models (such as OPT-66B). Furthermore, ConQuist provides significant advantages in lower-bit regimes, surpassing GPTQ and other baseline methods at 3-bit quantization by significantly reducing perplexity across various datasets for both GPT and BLOOM model families.

The key contributions of this study are summarised as follows:

- We are the first to apply the condition number as sensitivity metric to the layer-wise quantization of LLMs.

- Based on the condition number, we propose ConQuist, a novel mixed precision quantization framework for LLMs. ConQuist utilizes the condition number of layer activations to rank layers by sensitivity and allocate higher bits to layers with high sensitivity score.

- We present a novel perspective for analyzing performance degradation due to quantization through the lens of relative quantization output error. We additionally provide mathematical analysis deriving a theoretical bound that connects the variance of the relative quantization output error to the condition number of the input activations.

- ConQuist allocates bits to each layer based on the condition number of their activations, which is computed from a single calibration pass. Consequently, it does not require Hessian computations or any second-order information.

- Empirically, ConQuist can significantly recover the performance of quantized model and maintains performace almost same as full precision even on lower bit quantized models.

## 2 PRELIMINARY

### 2.1 POST TRAINING QUANTIZATION

Post-training quantization is increasingly recognized as an essential approach for the efficient deployment of large-scale models, substantially boosting inference speed and addressing many limitations of conventional high-precision computation. One of the most effective PTQ technique is GPTQ, GPTQ (Frantar et al., 2023) extends Optimal Brain Quantization (OBQ) (Frantar & Alistarh, 2022) by utilizing second-order optimization, which enables the derivation of a closed-form solution that reduces quantization error in theory. Furthermore, Activation-aware Weight Quantization (AWQ) (Lin et al., 2024) refines this process by introducing a scaling factor specifically aimed at minimizing the quantization error of significant weights. Nevertheless, these methods are primarily limited to weight quantization and do not address the reduction of activation bitwidths, thus missing out on further potential efficiency gains. SmoothQuant (Xiao et al., 2023) attempts to bridge this gap by empirically shifting the quantization burden from activations to weights through specialized transformations. Various technique that focus smoothning of activations helps dealing with outliers of activations such as ASER (Zhao et al., 2025). ASER proposes a low-rank activation smoothing and error reconstruction algorithm to compensate quantization loss. More recently, ABQ-LLM (Zeng et al., 2025) presents an arbitrary-bit quantization framework that supports flexible precision levels with superior inference performance.

## 3 PROPOSED METHODOLOGY

In this section, we formally introduce our proposed method, ConQuist, **Condition Number Aware Mixed Precision Quantization**. The central innovation of ConQuist lies in mixed precision quantization strategy based on the condition number of activations. Unlike traditional mixed precision quantization approaches, ConQuist begins by considering the entire layers of model and corresponding activation inputs, denoted as $X$. ConQuist computes condition number of input activations of

each layer and denote condition number as sensitivity score of that particular layer. The precision allocation is motivated by the fact that input activation exhibit low rank properties and contains outliers, leading to ill-conditioning of input activations. Our approach follows the widely adopted framework utilized in leading PTQ methods (Frantar et al., 2023), wherein each layer is quantized independently by formulating and solving a local reconstruction problem. This strategy allows for effective calibration while obviating the need for complete model retraining.

## 3.1 MATHEMATICAL FORMULATION:

Let a linear layer be represented as $Y = XW$, where $X \in \mathbb{R}^{n \times d}$ denotes the input activation matrix and $W \in \mathbb{R}^{d \times m}$ represents the full-precision weight matrix. The objective is to obtain a quantized weight matrix $\widehat{W}$ that closely approximates the behavior of the original layer by minimizing the squared Euclidean norm between the outputs of the full-precision and quantized weights:

$$\arg \min_{\widehat{W}} \left\| XW - X\widehat{W} \right\|_2^2 \tag{1}$$

To understand the causes of performance degradation in layer-wise post-training quantization, we analyze the impact of weight quantization within linear layers. In quantized models, the original weight matrix $W$ is substituted with its quantized counterpart $\widehat{W} = W + \Delta W$, where $\Delta W$ denotes the quantization error resulting from the discretization process. Substituting this into the linear operation gives:

$$\widehat{Y} = X\widehat{W} = X(W + \Delta W) = XW + X\Delta W. \tag{2}$$

Here, $XW$ represents the original output computed with full-precision weights, while the term $X\Delta W$ characterizes the output error attributable exclusively to weight quantization. This output error, denoted $\Delta Y$, can be expressed as:

$$\Delta Y = \widehat{Y} - Y = X\Delta W. \tag{3}$$

The degradation in model performance due to quantization is caused by the interaction between the input activations $X$ and the weight quantization $\Delta W$. Importantly, even if the magnitude of $\Delta W$ is constrained, the resulting error $\Delta Y$ can still be large if the input activation $X$ amplifies perturbations due to quantization especially for layers with ill-conditioned activations.

To quantify this effect, the euclidean norm of the output error is bounded as:

$$\|\Delta Y\|_2 = \|X\Delta W\|_2 \leq \|X\|_2 \|\Delta W\|_2, \tag{4}$$

where $\|X\|_2$ is the spectral norm (largest singular value) of the input matrix. This highlights that the sensitivity of the layer to weight quantization is directly influenced by the spectral properties of the input.

We now establish relationship between model sensitivity to quantization and the condition number of the input activations. Taking the lower bounds on the spectral norms of the output and its quantization, as follows:

$$\|Y\|_2 = \|XW\|_2 \geq \sigma_{\min}(X)\|W\|_2, \tag{5}$$

where $\| \cdot \|_2$ denotes the spectral norm (i.e., the operator 2-norm), and $\sigma_{\min}(W)$ is the smallest singular value of the weight matrix $W$.

By combining the inequalities (4) and (5), we derive an upper bound on the relative output distortion due to weight quantization, as:

$$\frac{\|\Delta Y\|_2}{\|Y\|_2} \leq \frac{\|X\|_2 \|\Delta W\|_2}{\sigma_{\min}(X)\|W\|_2} = \kappa(X) \cdot \frac{\|\Delta W\|_2}{\|W\|_2}, \tag{6}$$

where $\kappa(X) = \frac{\|X\|_2}{\sigma_{\min}(X)}$ is the condition number of the input activations. This formulation illustrates that the impact of quantization-induced perturbations is amplified in layers with poorly conditioned input activations. In other words, for a fixed quantization error $\|\Delta W\|_2$, a higher condition number of input activations leads to a proportionally larger output error.

**Theorem 1.** *(Widrow & Kollár, 2008) Let $X \in \mathbb{R}^{n \times d}$ be the input activation matrix, $W \in \mathbb{R}^{d \times m}$ the full-precision weight matrix, and $Y = XW \in \mathbb{R}^{n \times m}$ the output. Suppose the quantized weights are given by $\hat{W} = W + \Delta W$, where each entry of $\Delta W$ is independent and uniformly distributed on $\left[ -\frac{\Delta}{2}, \frac{\Delta}{2} \right]$, with $\Delta = R \cdot 2^{1-b}$ for a symmetric uniform $b$-bit quantizer over the range $[-R, R]$. Then Total Quantization error has varience*

$$\mathbb{E}[\Delta W^2] = dm\frac{\Delta^2}{12}.$$

**Theorem 2** (Bound on Relative Output Error from Weight Quantization). *Let $X \in \mathbb{R}^{n \times d}$ be the input activation matrix, $W \in \mathbb{R}^{d \times m}$ the full-precision weight matrix, and $Y = XW \in \mathbb{R}^{n \times m}$ the output. Suppose the quantized weights are given by $\hat{W} = W + \Delta W$, where $W$ follows uniform quantization, for a symmetric uniform $b$-bit quantizer over the range $[-R, R]$. Then the expected relative squared output error is bounded as:*

$$\frac{\mathbb{E}[\|\Delta Y\|_2^2]}{\|Y\|_2^2} \leq \frac{\alpha}{\|W\|_2^2} \cdot \kappa^2(X) \cdot 2^{-2b},$$

*where $\kappa(X) = \frac{\sigma_{\max}(X)}{\sigma_{\min}(X)}$ is the spectral condition number of $X$.*

*Proof.* Let the quantization error matrix be denoted by $\Delta W = \hat{W} - W$, and define the corresponding output error as:

$$\Delta Y = \hat{Y} - Y = X \Delta W.$$

By the submultiplicative property of the euclidean norm:

$$\|\Delta Y\|_2 = \|X \Delta W\|_2 \leq \|X\|_2 \cdot \|\Delta W\|_2 = \sigma_{\max}(X) \cdot \|\Delta W\|_2.$$

Taking expectation over the quantization error:

$$\mathbb{E}[\|\Delta Y\|_2^2] \leq \sigma_{\max}^2(X) \cdot \mathbb{E}[\|\Delta W\|_2^2].$$

From the uniform quantization model, each entry of $\Delta W \in \mathbb{R}^{d \times m}$ has variance $\mathbb{E}[\Delta w^2] = \frac{\Delta^2}{12}$, and thus:

$$\mathbb{E}[\|\Delta W\|_2^2] = dm \cdot \frac{\Delta^2}{12} = \alpha \cdot 2^{-2b}, \quad \text{where } \alpha = \frac{dm(2R)^2}{12}.$$

Hence,

$$\mathbb{E}[\|\Delta Y\|_2^2] \leq \sigma_{\max}^2(X) \cdot \alpha \cdot 2^{-2b}.$$

To express this as a relative error, we divide by $\|Y\|_2^2$:

$$\frac{\mathbb{E}[\|\Delta Y\|_2^2]}{\|Y\|_2^2} \leq \frac{\sigma_{\max}^2(X) \cdot \alpha \cdot 2^{-2b}}{\|Y\|_2^2}.$$

Using the assumption $\|Y\|_2 \geq \sigma_{\min}(X)\|W\|_2$, we obtain:

$$\|Y\|_2^2 \geq \sigma_{\min}^2(X) \cdot \|W\|_2^2.$$

Substituting into the denominator:

$$\frac{\mathbb{E}[\|\Delta Y\|_2^2]}{\|Y\|_2^2} \leq \frac{\sigma_{\max}^2(X)}{\sigma_{\min}^2(X)} \cdot \frac{\alpha \cdot 2^{-2b}}{\|W\|_2^2} = \kappa^2(X) \cdot \frac{\alpha}{\|W\|_2^2} \cdot 2^{-2b}.$$

$\square$

## 3.2 DISCUSSION

Mathematical formulation and aforementioned theoretical analysis reveals several important insights for the design and optimization of mixed precision quantization strategies in LLMs:

- **Sensitivity to conditioning.**
  The relative output error due to quantized weights scales proportionally with the square of the spectral condition number, $\kappa^2(X)$, of the input activation matrix. This indicates that layers with high condition numbers, commonly referred to as *ill-conditioned* are significantly more sensitive to quantization error. Even minor perturbations in weights can result in disproportionately large errors in the output. As a result, conditioning serves as a fundamental indicator of quantization robustness.

- **Limitations of uniform quantization.**
  The result highlights a critical weakness of conventional uniform quantization: it fails to account for the structural diversity in layer sensitivity. In deep models with multiple layers, some layers may be well-conditioned and robust to low-bit quantization, while others may require higher precision. By treating all layers equally, uniform quantization may either waste bits on robust layers or severely degrade performance on sensitive ones. Mixed precision quantization, where bit-widths vary across layers better aligns with the sensitivity profile of the network and enables more efficient hardware utilization.

- **Theoritical motivation for condition number aware bit allocation.**
  Since the expected output error is bounded by a factor of $\kappa^2(X) \cdot 2^{-2b}$, it follows that uniform quantization (i.e., using the same bit-width $b$ for all layers) may be suboptimal, especially in networks with heterogeneous conditioning across layers. A more effective strategy is to assign bit-widths adaptively, increasing $b$ for layers with higher $\kappa(X)$ to mitigate the risk of high quantization error.

### 3.3 ALGORITHM

Based on aforementioned findings, we propose ConQuist, a mixed precision quantization strategy that leverages the condition number of layer-wise activations to guide bit-width allocation across the network. Let $\kappa_i$ denote the condition number of the input activation matrix for the $i$-th layer, and $n_i$ represent the number of parameters in that layer. The condition number is computed as:

$$\kappa_i = \frac{\sigma_{\max}(X_i)}{\sigma_{\min}(X_i)}, \tag{7}$$

where $\sigma_{\max}(X_i)$ and $\sigma_{\min}(X_i)$ are the largest and smallest singular values of the input activation matrix $X_i$, respectively. Further we sort the layers in descending order, according to their activation condition numbers $\kappa_i$, and allocates bit-widths accordingly. Layers whose condition numbers lies in top 10% are assigned a higher bit-width, while those with lower sensitivity are quantized using fewer bits. This strategy ensures that precision is focused on the most sensitive layers, thereby reducing overall memory consumption without significantly degrading performance. To mitigate the computational overhead of condition number estimation, we approximate $\kappa_i$ by using random sampling algorithms from the randomized linear algebra literature (Rudelson & Vershynin, 2007; Mahoney et al., 2011).

## 4 EXPERIMENTS AND RESULTS

Our experiments are designed to address two core questions:

1. How does selectively increasing precision for the most sensitive layers affect quantized model behavior?

2. How does the ConQuist method balance memory consumption with the quality of model predictions?

To answer these, we rank all layers in each model by their computed sensitivity scores and select the top 10% as the most critical. These layers are quantized using one additional bit compared to the rest of the network, to improve robustness in regions most prone to quantization error.

### 4.1 MODELS AND DATASETS

We conduct our study across multiple widely used LLM architectures, including the OPT (125M, 350M, 1.3B, 2.7B, 6.7B, 13B, 30B, 66B) and BLOOM (560M, 1.1B, 1.7B, 3B, 7.1B) families.

OPT and BLOOM models are quantized and validated using the C4 (Raffel et al., 2020), Penn Treebank (Marcus et al., 1994), and WikiText2 (Merity et al., 2016) datasets. To demonstrate the superior performance of our approach, we compare ConQuist against several recent quantization baseline methods, including GPTQ, AWQ, ZQV2 (Yao et al., 2023), and AgileQ (Shen et al., 2024). To ensure a fair comparison, we adopt the experimental setup and dataset requirements consistent with those outlined in GPTQ. All details on experimental setup, calibration datasets are provided in supplementary material and to ensure reproducibility of the ConQuist, codes are also provided in supplementary material. We have provided extra experiments, ablation study and more details in Appendix A.2.

## 4.2 BASELINES AND METRICS

We compare ConQuist with two reference post-training quantization techniques: GPTQ and standard round-to-nearest quantization, each tested with 3-bit and 4-bit precision. For our approach, both "3+ (3.1 bit)" and "4+ (4.1 bit)" configurations are explored, where a minority of layers are assigned one extra bit based on sensitivity. Although this method slightly raises storage requirements, but typically leads to significantly lower perplexity. To ensure a consistent and unbiased assessment, all quantization settings (including sequence length and number of evaluation samples) mirror those used in prior GPTQ work. We report perplexity for each model/dataset combination.

Table 1: Perplexity of OPT on the WikiText2 dataset.

| OPT | Bits (W/A) | 125M | 1.3B | 2.7B | 6.7B | 13B | 30B |
|---|---|---|---|---|---|---|---|
| FULL | 16/16 | 27.65 | 14.63 | 12.47 | 10.86 | 10.13 | 9.56 |
| RTN | 4/16 | 37.28 | 48.18 | 16.92 | 12.1 | 11.32 | 10.98 |
| GPTQ | 4/16 | 31.12 | 15.47 | 12.87 | 11.39 | 10.31 | 9.63 |
| AWQ | 4/16 | 33.96 | 16.85 | 14.61 | 12.44 | 11.60 | 10.75 |
| ZQV2 | 4/16 | 36.71 | 19.38 | 17.92 | 11.91 | 10.67 | 10.10 |
| AgileQ | 8/8 | 31.52 | 15.90 | 13.43 | 11.43 | 10.42 | 9.70 |
| **ConQuist** | 4+/16 | **29.81** | **15.13** | **12.83** | **11.07** | **10.19** | **9.50** |
| GPTQ | 3/16 | 53.85 | 20.97 | 16.88 | 14.86 | 11.61 | 10.27 |
| **ConQuist** | 3+/16 | **40.44** | **17.34** | **14.428** | **11.52** | **10.66** | **9.86** |

Table 2: Perplexity of OPT on the C4 dataset.

| OPT | Bits (W/A) | 125M | 1.3B | 2.7B | 6.7B | 13B | 30B |
|---|---|---|---|---|---|---|---|
| FULL | 16/16 | 26.56 | 16.07 | 14.34 | 12.71 | 12.06 | 11.44 |
| RTN | 4/16 | 33.91 | 24.51 | 18.43 | 14.36 | 13.36 | 13.46 |
| GPTQ | 4/16 | 29.22 | 16.97 | 15.00 | 13.18 | 12.26 | 11.57 |
| ZQV2 | 4/16 | 30.92 | 17.93 | 18.32 | 13.01 | 12.07 | 11.33 |
| AgileQ | 8/8 | 28.43 | 16.72 | 14.91 | 12.70 | 11.77 | 11.14 |
| **ConQuist** | 4+/16 | **26.79** | **15.25** | **13.52** | **11.92** | **11.30** | **10.75** |
| GPTQ | 3/16 | 42.41 | 21.63 | 18.17 | 17.14 | 13.34 | 12.23 |
| **ConQuist** | 3+/16 | **34.34** | **17.79** | **15.00** | **12.65** | **11.75** | **11.05** |

## 4.3 MAIN RESULTS

We present the main results for the OPT and BLOOM model families in Tables 1–4. We demonstrate the results of OPT and Bloom Family in Table 1-4. In most of the case, we observe that ConQuist outperforms GPTQ (and correspondingly RTN) at similar model size by a significant margin, especially on smaller models. This improvement in perplexity results from ConQuist's ability to allocate higher precision to ill conditioned layers with high sensitivity thereby reducing quantization induced

output distortion while still achieving compression by using lower precision in well-conditioned layers. For a model with $N$ parameters, allocating 5 bits to 10% of the weights and 4 bits to the remaining 90% results in: $0.1 \times 5 + 0.9 \times 4 = 4.1$ bits per parameter, which is a $\sim 2.5\%$ increase over uniform 4-bit quantization, yet it yields significantly lower perplexity. Similarly, allocating 4 bits to 10% of the weights and 3 bits to the rest results in an average of: $0.1 \times 4 + 0.9 \times 3 = 3.1$ bits per parameter. This corresponds to only a $\sim 3.3\%$ increase over standard 3-bit quantization.

This small overhead can lead to noticeable improvements in model performance as illustrated in Figure 2. For example, on OPT-30B (C4), ConQuist achieves 10.75 perplexity at 4+ bits, outperforming GPTQ (11.57) and RTN (13.46). At 3+ bits, ConQuist reaches 11.05, reducing the gap to full precision by a larger margin than GPTQ. On BLOOM-7.1B, it improves over GPTQ by more than 1.3 points at 4 bit and 2 points at 3-bit. In most cases, the perplexity reduction achieved by ConQuist over GPTQ matches or exceeds the improvement GPTQ offers over RTN. Notably, at 4+ bit precision, ConQuist closes the perplexity gap to full precision by nearly twice the margin compared to GPTQ. For instance, on OPT-1.3B (C4), ConQuist lowers perplexity from 16.97 (GPTQ) to 15.25, surpassing GPTQ by 1.72 points. This improvement is nearly twice the gain GPTQ achieves over RTN ($24.51 \rightarrow 16.97$). These results underscore ConQuist's precision efficiency, it closes a larger portion of the gap to full precision (16.07) than prior methods, confirming its superior ability to preserve model performance under low-bit quantization. Furthermore, we report the performance of the proposed approach, ConQuist, on the PTB dataset using the OPT and BLOOM model families in Appendix A.3.

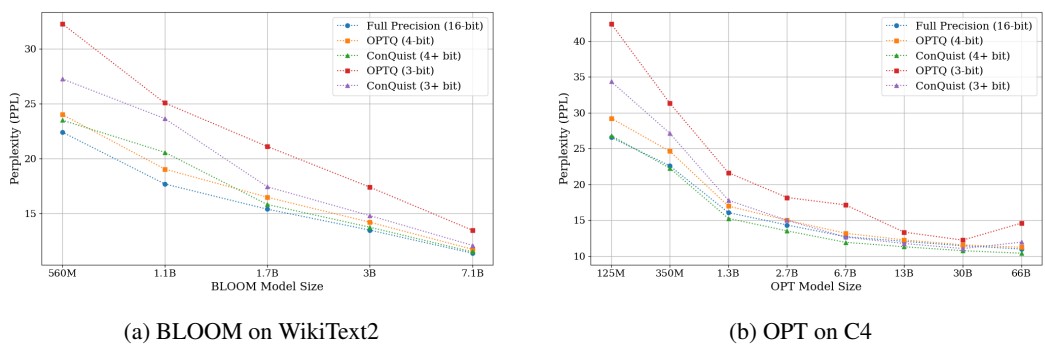

(a) BLOOM on WikiText2            (b) OPT on C4

Figure 2: Perplexity comparison of BLOOM and OPT models quantized with ConQuist.

Table 3: Perplexity of BLOOM on the WikiText2 dataset.

| BLOOM | Bits (W/A) | 560M | 1.1B | 1.7B | 3B | 7.1B |
|---|---|---|---|---|---|---|
| FULL | 16/16 | 22.42 | 17.69 | 15.39 | 13.48 | 11.37 |
| RTN | 4/16 | 25.90 | 22 | 16.97 | 14.76 | 12.1 |
| GPTQ | 4/16 | 24.03 | 19.05 | 16.48 | 14.2 | 11.73 |
| ZQV2 | 4/16 | 25.31 | 23.90 | 16.93 | 14.65 | 12.06 |
| AgileQ | 8/8 | 24.01 | **18.82** | 16.23 | 14.05 | 11.73 |
| **ConQuist** | 4+/16 | **23.51** | 20.57 | **15.81** | **13.74** | **11.49** |
| GPTQ | 3/16 | 32.31 | 25.08 | 21.11 | 17.40 | 13.47 |
| **ConQuist** | 3+/16 | **27.27** | **23.65** | **17.43** | **14.801** | **12.08** |

### 4.3.1 FURTHER ANALYSIS

To assess the generality of our method, we integrate ConQuist into an alternative quantization framework: SmoothQuant. In its default configuration, SmoothQuant quantizes both weights and activations to 8 bits. To test compatibility, we apply ConQuist by reducing weight precision to 6 bits for 50% of the layers, guided by sensitivity scores, while keeping activation precision fixed at 8 bits. This yields an average of 7 bits per weight, corresponding to a 12.5% reduction in weight storage.

Table 4: Perplexity of BLOOM on the C4 dataset.

| BLOOM | Bits (W/A) | 560M | 1.1B | 1.7B | 3B | 7.1B |
|---|---|---|---|---|---|---|
| FULL | 16/16 | 26.60 | 22.05 | 19.49 | 17.49 | 15.2 |
| RTN | 4/16 | 29.89 | 24.44 | 21.26 | 18.76 | 16.06 |
| GPTQ | 4/16 | 28.00 | 23.25 | 20.55 | 18.10 | 15.60 |
| ZQV2 | 4/16 | 27.10 | 25.99 | 19.47 | 17.26 | 14.83 |
| AgileQ | 4/16 | 26.39 | **21.80** | 29.18 | 16.96 | 14.70 |
| **ConQuist** | 4+/16 | **25.37** | 23.63 | **18.60** | **16.43** | **14.28** |
| GPTQ | 3/16 | 35.78 | **28.83** | 25.34 | 21.25 | 17.67 |
| **ConQuist** | 3+/16 | **33.04** | 28.94 | **21.96** | **18.68** | **15.68** |

As shown in Table 5, this modification leads to a significant reduction in model size with only a small increase in perplexity. For example, on OPT-13B, perplexity increases from 10.31 to 11.08 despite halving weight precision in half the layers. Similar trends hold across other model sizes.

These results demonstrate that ConQuist can be applied on top of other quantization schemes such as SmoothQuant, enabling additional compression with minimal impact on performance. This further confirms the flexibility of our method in finding optimized trade-offs between memory efficiency and perplexity across different quantization backbones.

Table 5: ConQuist applied to SmoothQuant with 6+ bits quantization for weights

| OPT | Bits (W/A) | 1.3B | 2.7B | 6.7B | 13B |
|---|---|---|---|---|---|
| SmoothQuant | 8/8 | 14.77 | 12.46 | 10.88 | 10.31 |
| **ConQuist** | 6+/8 | 15.17 | 13.05 | 11.48 | 11.08 |

## 5 CONCLUSION

In this paper, we present ConQuist, a novel mixed precision quantization framework for large language models, utilizing the condition number of layer-wise activations as a sensitivity metric. ConQuist systematically identifies and ranks the sensitivity scores of layers according to the condition number of input activations. ConQuist assigns higher precision only to a small subset of highly sensitive layers, while quantizing the remaining layers at lower precision. Our empirical results demonstrate that this targeted approach of ConQuist not only outperforms existing baseline quantization algorithms but also can be readily incorporated into any post-training quantization method to achieve mixed precision quantization. ConQuist highlights that significant performance gains can be attained by judiciously allocating higher bit-widths to a limited number of sensitive layers, thereby advancing the efficiency and effectiveness of quantized large language models.

## 6 FUTURE WORK & LIMITATION

We introduce the first quantization method with condition number based relative error bounds, opening promising directions for future advancements. Despite the strong performance of ConQuist, a few limitations remain. At present, ConQuist uses fixed method as top $10\%$ for condition number based higher bit assignment, though effective, could be enhanced by adaptive or learnable policies that respond to model specific sensitivity profiles. Moreover, this work focuses solely on weight quantization, extending ConQuist to jointly optimize both weights and activations (on smaller bits *e.g.* 2 or 3 bits) under the same sensitivity aware strategy presents a promising direction for further improving compression performance trade-offs.

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

# A APPENDIX

## A.1 RELATED WORK

### A.1.1 CONDITION NUMBER

The condition number, a foundational concept in numerical analysis of matrix (Golub & Van Loan, 2013; Turing, 1948), measures how sensitive a function's output is to small changes in its input. A matrix with a large condition number is referred to as ill-conditioned, indicating that even minor input perturbations can lead to substantial changes in the output. This property presents significant challenges for achieving numerical stability and reliability.

### A.1.2 MIXED PRECISION QUANTIZATION

Mixed precision quantization is a technique where different parts of a large language model, such as weights, activations, or entire layers are assigned varying bit-widths based on their sensitivity or importance, enabling significant memory and computational efficiency while retaining high model accuracy. Mixed precision quantization of LLMs is an active area of research. Recently proposed KVTuner (Li et al., 2025b) studies sensitivity of key-value cache and their role in model quantization, KVTuner allocate different bits of key or value, based on their sensitivity. Moreover, ResQ (Saxena et al., 2024), uses uses low-rank projections for mixed precision mapping across all model component. MixQ (Chen et al., 2024b) advances bit assignment by making it outlier-aware to further boost low-bit quantization robustness. Mixture of Quantization-Aware Experts (MoQAE) (Tao et al., 2025), uses Uses chunk-based processing for bit allocation. In MoQAE, each chunk is routed through a lightweight multilayer perceptron (MLP) that selects the most suitable precision of bits, adapting dynamically per chunk. Exploiting LLM Quantization (Egashira et al., 2024) demonstrates the superiority of mixed precision over uniform int8 quantization. Collectively, these methods highlight the rapid progression toward highly adaptive, practical mixed precision quantization for scalable and efficient large language model deployment.

## A.2 ABLATION STUDY

The ConQuist allocation strategy raises the question of how many layers truly benefit from higher precision. To answer this, we conduct a controlled experiment where we vary the ratio of layers assigned one extra bit, guided by sensitivity scores. All settings maintain the same 4-bit base precision, and the only change is the proportion of layers boosted to 5 bits. We begin with the 10% allocation used throughout our main results and progressively increase it to 20% and 30%. As shown in Table 6, the perplexity improves marginally. For example, from 15.25 to 15.20 on OPT-1.3B despite a $3\times$

increase in the number of boosted layers. A similar trend holds across larger models. This result suggests that most performance gains are concentrated in a small set of highly sensitive layers. Once those are handled, additional precision offers diminishing returns. This aligns with our sensitivity analysis (see Figure 1), where only a few layers exhibit significantly higher condition numbers.

Table 6: Ablation study of different bit allocation ratios across layers of OPT models, guided by sensitivity scores

| OPT | Bits ratio | 1.3B | 2.7B | 6.7B | 13B |
|-----|-----------|------|------|------|------|
| | 4+/16 | **15.25** | **13.52** | **11.92** | **11.3** |
| **ConQuist** | 4+ (0.2:0.8) | **15.23** | **13.48** | **11.92** | **11.28** |
| | 4+ (0.3:0.7) | **15.20** | **13.47** | **11.93** | **11.29** |

## A.3 RESULTS ON PTB DATASETS ON OPT AND BLOOM MODELS

Table 7: Perplexity of OPT on the PTB dataset.

| OPT | Bits (W/A) | 125M | 350M | 1.3B | 2.7B | 6.7B | 13B | 30B | 66B |
|-----|-----------|------|------|------|------|------|-----|-----|-----|
| Full | 16/16 | 38.99 | 31.08 | 20.29 | 17.97 | 15.52 | 14.04 | 13.36 | 12.01 |
| RTN | 4/16 | 53.89 | 36.79 | 57.30 | 31.05 | 18.84 | 16.51 | 15.4 | 225.66 |
| GPTQ | 4/16 | 45.17 | 34.52 | 21.85 | 19.14 | 16.56 | 14.94 | 14.26 | 13.81 |
| **ConQuist** | 4+/16 | **35.15** | **27.85** | **17.40** | **15.39** | **13.23** | **12.44** | **11.88** | **11.53** |
| GPTQ | 3/16 | 73.19 | 47.08 | 32.1 | 24.81 | 21.88 | 16.68 | 15.36 | 28.12 |
| **ConQuist** | 3+/16 | **43.35** | **32.67** | **19.43** | **16.49** | **13.80** | **12.77** | **12.12** | **12.63** |

Table 8: Perplexity of BLOOM on the PTB dataset.

| BLOOM | Bits (W/A) | 560M | 1.1B | 1.7B | 3B | 7.1B |
|-------|-----------|------|------|------|-----|------|
| Full | 16/16 | 43.69 | 57.96 | 30 | 25.34 | 20.83 |
| RTN | 4/16 | 51.10 | 66.85 | 33.58 | 27.68 | 22.42 |
| GPTQ | 4/16 | 46.97 | 62.47 | 31.84 | 26.49 | 21.67 |
| **ConQuist** | 4+/16 | **42.58** | **60.87** | **28.42** | **23.51** | **19.6** |
| GPTQ | 3/16 | 70.35 | 87.04 | 46.11 | 34.02 | 26.14 |
| **ConQuist** | 3+/16 | **50.01** | **68.18** | **31.17** | **25.08** | **20.44** |

