# OpenReview forum: "ConQuist: Condition Number Aware Quantization for LLMs"
_ICLR.cc/2026/Conference — ICLR 2026 Conference Withdrawn Submission_

### Official Review · Reviewer_f8Jh · 2025-10-29

**Soundness:** 3
**Presentation:** 2
**Contribution:** 2
**Rating:** 4
**Confidence:** 4

**Summary:**

This paper introduces a novel quantization sensitivity metric based on the derivation that high condition numbers are prone to high quantization error. And then ConQuist is proposed to allocate higher bitwidth to layers with high condition numbers and lower bitwidth to layers with low condition numbers.

**Strengths:**

1. This paper proposes a novel salience metric considering the impact of activation perturbation, which means that in addition to the popular Hessian metric, other inherent factors influencing quantization performance should be further explored.
2. ConQuist outperforms various baselines and achieves great performance under W3A16 and W4A16.

**Weaknesses:**

1. Although this salience metric is proposed for the first time, the idea based on input activations is not novel, where similar analyses have been explored in AWQ [1], OWQ [2], etc.
2. Although the new metric appears to be effective, it remains to be explored whether it can truly replace the more commonly used Hessian-based metrics. However, the authors ignores to discuss it.
3. The evaluated LLMs in this paper are actually out of date. The performance on more widely used LLaMA3 and Qwen2 are ignored.

>[1] Lin J, Tang J, Tang H, et al. Awq: Activation-aware weight quantization for on-device llm compression and acceleration[J]. Proceedings of machine learning and systems, 2024, 6: 87-100.

>[2] Lee C, Jin J, Kim T, et al. Owq: Outlier-aware weight quantization for efficient fine-tuning and inference of large language models[C]//Proceedings of the AAAI Conference on Artificial Intelligence. 2024, 38(12): 13355-13364.

**Questions:**

1. More sensitivity visualization on LLMs with more widely used structures (LLaMA, Qwen) should be provided to demonstrate the universality of the phenomenon.
2. In Table 1-3, the comparison should be focus on weight-only baselines. AgileQ (W8A8) is unnecessary.
3. The combination of the proposed sensitivity metric with other baselines with Hessian-based metric should be explored, such as PBLLM [3], VPTQ [4] and BiLLM [5], to demonstrate its consistent superiority.

>[3] Shang Y, Yuan Z, Wu Q, et al. Pb-llm: Partially binarized large language models[J]. arXiv preprint arXiv:2310.00034, 2023.

>[4] Liu Y, Wen J, Wang Y, et al. Vptq: Extreme low-bit vector post-training quantization for large language models[J]. arXiv preprint arXiv:2409.17066, 2024.

>[5] Huang W, Liu Y, Qin H, et al. Billm: Pushing the limit of post-training quantization for llms[J]. arXiv preprint arXiv:2402.04291, 2024.

---

### Official Review · Reviewer_Rn6X · 2025-10-30

**Soundness:** 2
**Presentation:** 2
**Contribution:** 2
**Rating:** 2
**Confidence:** 3

**Summary:**

The paper introduces ConQuist, a novel post-training quantization (PTQ) method designed for large language models (LLMs). It aims to mitigate quantization errors by leveraging the condition number of layer activations, which measures the sensitivity of a layer's output to small perturbations. ConQuist applies mixed precision quantization, allocating higher precision (e.g., 5-bit) to layers with higher condition numbers and 4-bit precision to less sensitive layers. The approach avoids the computational burden of second-order information, as it only requires activation data, making it efficient and scalable. Through empirical results, the authors demonstrate that ConQuist outperforms baseline PTQ methods, reducing perplexity by up to 20% across several benchmarks, even at lower bit-widths like 3-bit.

**Strengths:**

1. One of the strengths of the paper is the detailed theoretical derivation, which feels rigorous and well-supported. The approach is backed by solid mathematical analysis, providing a strong theoretical foundation for the proposed method, and there don't seem to be any significant issues with the reasoning or formulation.
2. The method used to assess layer sensitivity is computationally efficient, with minimal overhead and fast execution. This makes the approach scalable, especially for large models, without introducing significant delays in the quantization process.

**Weaknesses:**

1. The experimental models used are relatively outdated. There is a lack of support for newer model architectures, such as LLaMA, Qwen, or models with MoE structures. Including these more recent architectures would help demonstrate the generalizability and relevance of the proposed method in current cutting-edge model designs.
2. A notable drawback of the paper is the lack of demonstration of acceleration effects. While the method shows improvements in model performance, it would be valuable to showcase how ConQuist impacts inference speed or overall computational efficiency, particularly in terms of reducing latency and resource usage during deployment.
3. The baseline comparisons are somewhat limited, lacking comparisons with advanced quantization methods such as QuaRot and other state-of-the-art mixed-precision techniques. Including these methods would provide a more comprehensive evaluation of ConQuist’s performance and demonstrate its competitive edge against the latest advancements in the field.

**Questions:**

See Weaknesses

---

### Official Review · Reviewer_nwdH · 2025-10-31

**Soundness:** 2
**Presentation:** 3
**Contribution:** 2
**Rating:** 2
**Confidence:** 4

**Summary:**

This paper introduces a post-training quantization (PTQ) technique named ConQuist, which is designed for large language models (LLMs). ConQuist leverages the condition number of layer activations to guide the allocation of quantization bitwidths, enabling mixed-precision quantization. Through mathematical analysis, the authors demonstrate that layers with larger activation condition numbers are more sensitive to quantization errors, and thus propose assigning higher quantization precision to such layers.

**Strengths:**

(1) ConQuist employs the spectral condition number of layer-wise activations as a sensitivity metric to allocate bitwidths, which provides an interesting perspective for weight-only quantization.

**Weaknesses:**

(1) The authors only report perplexity results, which makes it difficult to comprehensively evaluate ConQuist’s capabilities. It is recommended that the authors conduct further evaluations on a broader range of downstream tasks, including zero-shot accuracy benchmarks (e.g., Lambada, Hellaswag, ARC-Challenge) and code generation tasks (e.g., HumanEval, MBPP).
(2) The large language models (LLMs) used in this paper—OPT and BLOOM—were released in 2022. No evaluations have been performed on more recent open-source models, such as the Llama 3 and Qwen 3 series. Extending experiments to these newer models would better demonstrate ConQuist’s generalizability.
(3) Practical deployment details are lacking. For example, how are the 5-bit weights stored? Clarifying such implementation-specific details is essential for assessing the method’s practical applicability.
(4) There is a lack of efficiency-related evaluations. Key metrics like inference latency, throughput, and peak memory consumption—critical for evaluating a quantization method’s real-world utility—should be provided and compared against baselines.

**Questions:**

(1) Could the authors discuss more diverse bit combination strategies (e.g., 3+4+5) by incorporating the condition number? The paper has already presented results for the 3+ combination, which demonstrates the feasibility of this line of thinking, making further exploration of such strategies valuable.
(2) Could the authors provide comparisons of runtime efficiency against baselines such as AWQ and TensorRT-W4A16? Key metrics for comparison should include end-to-end throughput and peak memory usage, as these are critical for evaluating practical applicability.
(3) It is recommended that the authors present evaluation results on additional benchmarks, particularly zero-shot accuracy. This would enable a more comprehensive assessment of ConQuist’s capabilities beyond the current experimental scope.
(4) For matrices with high sensitive scores, why are they not retained with higher bitwidths (e.g., 6-bit)? Additionally, the authors should clarify the trade-off between using higher precision (for better performance) and efficiency (e.g., speed, memory overhead) in such cases.

---

### Official Review · Reviewer_MAe3 · 2025-11-06

**Soundness:** 1
**Presentation:** 2
**Contribution:** 1
**Rating:** 2
**Confidence:** 5

**Summary:**

ConQuist uses condition number κ(X) of activation matrices as a sensitivity metric for mixed-precision PTQ, allocating 5-bit to top 10% sensitive layers and 4-bit to others. Shows improvements over GPTQ/AWQ on OPT/BLOOM families, but suffers from mathematical errors, questionable novelty, and incomplete evaluation.

**Strengths:**

Condition number naturally captures error amplification in ill-conditioned activations. Equation (6) provides clear scaling relationship.

Table 5 shows integration with SmoothQuant, suggesting orthogonality to smoothquant.

**Weaknesses:**

Proof uses invalid inequality ∥Y∥₂ ≥ σ_min(X)∥W∥₂ (should involve Frobenius norm). The assumption fails precisely when X is ill-conditioned (σ_min ≈ 0), undermining the entire theoretical justification. No tightness analysis provided.

Missing comparisons with 2024-2025 SOTA: ResQ, LQER, IMPQ, KVTuner, QuaRot, SpinQuant, etc. Only compares against 2022-2023 uniform quantization methods (GPTQ, AWQ).

CondiQuant (Feb 2025) already explores condition numbers for quantization - direct competitor not discussed. This could be a huge misconduction.

**Questions:**

See weaknesses.

---

### Note · Authors · 2025-12-28

I have read and agree with the venue's withdrawal policy on behalf of myself and my co-authors.